# Endoscopic Diagnosis and Therapy for Epstein–Barr Virus-Associated Gastric Cancer

**DOI:** 10.3390/microorganisms11112619

**Published:** 2023-10-24

**Authors:** Hideo Yanai, Junko Fujiwara, Eiichiro Toyama, Hiroshi Okuda, Osamu Miura, Seiji Kaino, Jun Nishikawa

**Affiliations:** 1Department of Clinical Research, Hofu Institute of Gastroenterology, 14-33 Ekiminami, Hofu 747-0801, Yamaguchi, Japan; 2Department of Gastroenterology & Hepatology, Hofu Institute of Gastroenterology, 14-33 Ekiminami, Hofu 747-0801, Yamaguchi, Japan; 3Department of Surgery, Hofu Institute of Gastroenterology, 14-33 Ekiminami, Hofu 747-0801, Yamaguchi, Japan; 4Department of Clinical Research, National Hospital Organization Kanmon Medical Center, 1-1 Sotoura, Chofu, Shimonoseki 752-8510, Yamaguchi, Japan; 5Department of Laboratory Science, Yamaguchi University Graduate School of Medicine, 1-1-1Minamikogushi, Ube 755-8505, Yamaguchi, Japan

**Keywords:** carcinoma with lymphoid stroma (CLS), Epstein–Barr virus (EBV), Epstein–Barr-virus-associated gastric cancer, endoscopic diagnosis, endoscopic submucosal dissection (ESD), endoscopic therapy, *Helicobacter pylori*

## Abstract

Epstein-Barr-virus-associated gastric cancer (EBVaGC) represents almost 7% of all GC and is a distinct subtype of GC with extreme DNA hypermethylation. EBVaGC is a tumor-infiltrating lymphocyte-rich tumor with little lymph-node metastasis in its early stage and with a relatively favorable prognosis in its advanced stage. Using upper gastrointestinal endoscopy, we recognize EBVaGC as a mainly depressed type with SMT-like protrusion in the upper part of the stomach near the gastric mucosal atrophic border or remnant stomach. The EBVaGC recognition rate of 21.4% with the endoscopic motif is not high, and further progress in endoscopic diagnosis of EBVaGC is needed. As less invasive endoscopic therapy, the extension of the criteria of endoscopic submucosal dissection (ESD) for early EBVaGC with little lymph-node metastasis should be discussed. Endoscopic diagnosis of EBVaGC may be relevant for the selection of patients who could benefit from endoscopic treatment or chemotherapy.

## 1. Introduction

Gastric cancer (GC) is a major health problem worldwide. Recently, Epstein–Barr virus (EBV) association has been detected in almost 7% of all cases of GC. However, the specific clinical diagnosis and therapy for EBV-associated GC (EBVaGC) is not well developed. So, research on endoscopic diagnosis and therapy of EBVaGC is required.

Burke et al. reported the first case of EBV-positive GC identified using polymerase chain reaction in 1990 [1]. Thereafter, EBV-encoded small RNA1 (EBER-1) in situ hybridization (ISH) has been used as the standard method for the confirmation of EBVaGC [2]. Monoclonal EBV presents in the nuclei of all cancer cells of each EBVaGC lesion, and a causal role of EBV in the carcinogenesis of EBVaGC is suspected [3].

The Cancer Genome Atlas Network (TCGA) reported that EBVaGC is a distinct subtype of GC with extreme DNA hypermethylation and overexpression of programmed cell death 1 ligand 1 (PD-L1) and ligand 2 (PD-L2) [4]. EBVaGC has a low risk of lymph node metastasis (LNM) in the early stage and a relatively favorable prognosis in the advanced stage. Thus, a favorable result of endoscopic submucosal dissection (ESD) for early EBVaGC and chemotherapy including immune checkpoint inhibitors for advanced EBVaGC is expected [5,6,7,8].

Endoscopic diagnosis for EBVaGC will play an important role in the choice of therapeutic modality against EBVaGC. In this communication, we introduce the present status of endoscopic diagnosis and therapy for EBVaGC, including our clinical experience.

Our study was approved by the Ethics Committee of National Hospital Organization Kanmon Medical Center (H2801-2, 8 January 2016; H2909-2, 8 September 2017). Written informed consent was obtained from all participants.

## 2. Clinical Features of EBVaGC

In 1997, using EBER1-ISH, we identified 12 EBVaGC lesions among 124 GC lesions (9.7%) and reported their endoscopic and pathologic features [9]. From a study of the remnant stomach, EBV association was detected in 7 of 17 GC lesions (41.8%) [10]. EBVaGC lesions were characteristically mainly poorly differentiated, tumor-infiltrating lymphocyte-rich adenocarcinoma (carcinoma with lymphoid stroma, CLS). Many EBVaGC lesions were located in the upper part of the stomach. Endoscopically, EBVaGC lesions were mainly the depressed type, and some of them had a submucosal tumor (SMT)-like protrusion consisting of a CLS mass. The CLS nodule can be observed as a hypoechoic mass using endoscopic ultrasonography (EUS) [9,11]. Later, Yanagi et al. analyzed 80 lesions of EBVaGC (7.1%) found among 1067 resected GC cases. Patients with EBVaGC were relatively younger and predominantly male, and the lesions were located in the upper part of the stomach or remnant stomach. Histologically, CLS is characteristic for EBVaGC, but also there are some cases of EBVaGC with the ordinary differentiated type [12]. Recently, Hirabayashi et al. performed a meta-analysis of over 68,000 cases of GC and reported 7.5% positivity for EBVaGC [13].

The latent infection status of EBV is reported as latency I, expressing EBV-determined nuclear antigen (EBNA) 1 only, and is hardly detectable by cytotoxic T cells [3]. However, the CLS of the EBVaGC lesion is tumor-infiltrating lymphocyte (TIL) rich [14]. For advanced GC cases, Song et al. reported that patients with EBVaGC have a longer survival time and that the prognosis of EBVaGC depends on the patient’s cellular immune response [15].

For the tumor, overexpression of PD-L1/L2 in EBVaGC lesions may play some role in tumor cell survival. Kim et al. reported that, in patients with microsatellite instability-high and EBV-positive tumors, dramatic responses to immune-checkpoint targeted therapy were observed. Thus, further clinical trials of immune-checkpoint targeted therapy for advanced EBVaGC are warranted [16]. Furthermore, the potential immunological role of regulatory T cells in EBVaGC should be analyzed for future treatment strategies [17].

## 3. Location of EBVaGC and *Helicobacter pylori* (Hp)-Related Gastritis

Yanagi et al. reported that 87.5% of EGVaGC was located in the upper to middle part of the stomach [12]. Song et al. also reported that 83.8% of EBVaGC was located there, which was significantly different from the lower-part-predominant EBV-negative GC [15]. The large meta-analysis of Hirabayashi et al. is in accord with these reports [13].

In our experience, EBVaGC is located not only in the upper part of the stomach but also near the gastric mucosal atrophic border front of Helicobacter pylori (Hp)-related gastritis [18]. There are histologic Hp-related chronic active gastritis and mucosal atrophic changes near the EBVaGC lesions [14,19]. Camargo et al. reported that anti-Hp antibody positivity was 95% in EBVaGCs [20]. Thus, EBVaGC is thought to be a characteristic part of Hp-related GC located near the gastric mucosal atrophic border in the upper part of the stomach.

Gastric mucosal epithelial cells are negative for CD21, the infection receptor for EBV. However, Imai et al. reported that cell-to-cell contact was an efficient mode of EBV infection from an EBV-positive B cell line to a gastric epithelial cell line [21]. In clinical cases, EBV is detectable from non-cancerous gastric mucosal biopsy specimens from the gastric mucosa of Hp-related gastritis using PCR [22]. Considering these findings, we speculate that Hp-related gastritis may provide the microenvironment for cell-to-cell EBV infection from EBV-positive B cells to gastric epithelial cells near the atrophic border. Fukayama et al. named the molecular process of EBVaGC carcinogenesis from Hp-related gastritis the “gastritis-infection-cancer sequence of Epstein–Barr-virus-associated gastric cancer” [23].

## 4. Endoscopic Diagnosis of EBVaGC

For endoscopic diagnosis of EBVaGC, the endoscopic motif of the location in the upper part of the stomach or remnant stomach and the shape of the mainly depressed type with an SMT-like protrusion of the lesion is important (Figure 1). The histological heterogeneity of EBVaGC that consists of differentiated-type adenocarcinoma in the superficial portion and CLS in the deeper portion should be kept in mind.

Based on recent therapeutic trends, less invasive ESD and immune-checkpoint inhibitor therapy for GC, the clinical impact of EBV detection in GC lesions appears to be important. In the clinical setting, however, EBV testing for all GC lesions is not practical. Therefore, we have already reported a retrospective analysis of endoscopic and pathologic motifs in GC lesions in an attempt to determine criteria to narrow the number of candidates for EBV testing [24]. These EBV tests were ordered in the real-time clinical setting by endoscopists or by pathologists based on the pathologic motif of CLS. The EBER1-positive rate was 42.3% (11/26) by the endoscopic or pathologic motif. Endoscopists ordered EBV tests for 18 lesions with the endoscopic motif. EBV was negative for four lesions of non-GC (gastric endocrine cell carcinoma, gastric hepatoid carcinoma, gastric T-cell lymphoma, and gastritis of the remnant stomach). Of the remaining 14 lesions of GC, three (21.4%) were EBVaGC. Our EBVaGC rate of 21.4% with the endoscopic motif was not high. It is clear that further progress in endoscopic diagnosis of EBVaGC is needed. A combination of the endoscopic and pathologic motifs seems to be useful for the diagnosis of EBVaGC.

## 5. Endoscopic Therapy against EBVaGC

Currently, less-invasive ESD is widely used for early GC worldwide. However, poorly differentiated adenocarcinoma lesions of GC with invasion into the submucosal layer are an indication for surgical operation with lymph node dissection because of the risk of LNM. Early EBVaGC is believed to have a low risk of LNM. We have already reported a case of early EBVaGC treated with ESD [7] (Figure 2). Further extension of the criteria for early EBVaGC should be discussed [25]. In many cases, EBVaGC is located in the upper part of the stomach and remnant stomach. Such extension of the ESD criteria may prevent possible excessive surgery such as total gastrectomy.

Near-infrared photoimmunotherapy (PIT) is a newly developed therapeutic method for some advanced-stage cancers. In PIT against GC, near-infrared light may be administered to the surface of the GC lesion via an optical fiber diffuser through the endoscope. EBVaGC is a TIL-rich immunogenic tumor. Thus, future application of PIT for advanced EBVaGC as an endoscopic therapy is promising [26].

## 6. Conclusions

EBVaGC is a TIL-rich tumor with little lymph-node metastasis in its early stage and with a relatively favorable prognosis in its advanced stage. Using upper gastrointestinal endoscopy, we recognize EBVaGC as a mainly depressed type with SMT-like protrusions in the upper part of the stomach near the gastric mucosal atrophic border or remnant stomach. Clinical diagnosis of EBVaGC may be relevant for the selection of patients who could benefit from less invasive ESD or chemotherapy. Further clinical research in the field of surgery and chemotherapy is required.

## Figures and Tables

**Figure 1 microorganisms-11-02619-f001:**
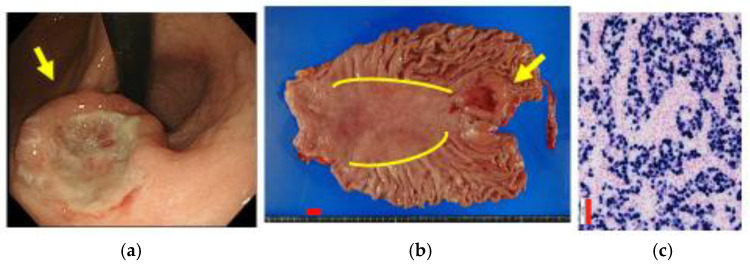
A case of advanced EBVaGC, 71 year-old female (Case from ref. [24]). (**a**) Endoscopic features. The GC lesion was mainly the depressed type with an SMT-like protrusion (yellow arrow) located near the gastric cardia. (**b**) Macroscopic view of the specimen of total gastrectomy. The GC lesion (yellow arrow) located near the gastric mucosal atrophic border (yellow line) of the upper part of the stomach. Hp-positive in non-cancer area. There is no lymph node metastasis. Red scale bar: ten millimeters. (**c**) Microscopic view of EBER-1 ISH. All tumor cell nuclei are EBER-1 positive in the CLS. Red scale bar: 50 μm.

**Figure 2 microorganisms-11-02619-f002:**
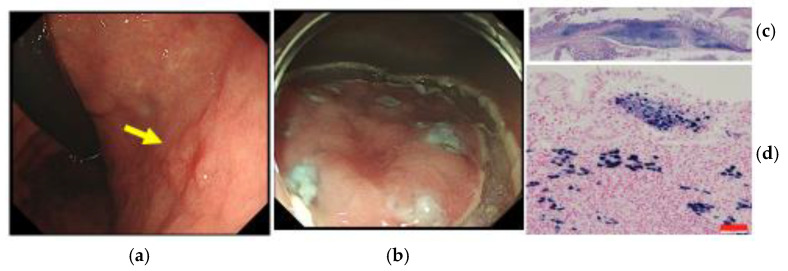
A case of early EBVaGC, 73 year-old male (case from ref. [7]). The present cancer lesion was his second lesion of early GC. He was once Hp positive and it had been eradicated. Seven months after the former endoscopic resection of the first non-EBV early GC lesion, another early GC lesion was detected with follow-up endoscopy. (**a**) Ordinary endoscopic features. A superficial depressed-type early gastric cancer lesion (yellow arrow) is located in the middle of the gastric body. (**b**) His early GC lesion treated with en block resection using ESD. (**c**) Low-power microscopic view of the ESD specimen with H&E stain. The tumor invaded 0.8 mm into the submucosal layer. The pathological type was CLS and lymphovascular infiltration was negative. (**d**) High-power microscopic view of the ESD specimen with EBER-1 ISH. All tumor cell nuclei are positive for EBER-1 ISH. The lesion was confirmed to be EBVaGC. All cell nuclei of TIL and non-cancer gastric epithelial cells are EBER-1 negative. Red scale bar: 50 μm. We recommended him to have an additional curative surgical operation with lymph node dissection based on the current Japanese gastric cancer treatment guidelines. The patient decided to not undergo the surgery or adjuvant chemotherapy. He is in good health at over six years since ESD without disease recurrence observed using endoscopy and CT. Some cases of early EBVaGC with invasion into the submucosal layer may be possible candidates for further extension of the ESD criteria. Further accumulation of outcome data on early EBVaGC treated with less invasive ESD is required to prevent possible overtreatment.

## Data Availability

There is no new data in this communication.

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
