# Peer review of "Endoscopic Diagnosis and Therapy for Epstein–Barr Virus-Associated Gastric Cancer"

_microorganisms, 2023, doi:10.3390/microorganisms11112619_

Round 1

Reviewer 1 Report

The communication manuscript is well written, and Epstein-Barr virus-associated gastric cancer (EBVaGC) is an essential topic to consider and interesting to readers. However, here are my comments to be considered: 

1. In the introduction, please highlight the gap in the literature and how this study can help to cover that. Please add a section on the study objectives. 

2. Please write a selection of the strengths and weaknesses of the study with future directions. 

Author Response

Dear Sir,

Our responses to the Reviewer’s comments are as follows.

To the Reviewer 1, 

We added the following sentence as suggested

1   ‘Gastric cancer (GC) is one of major health problems throughout the world. Recently, Epstein-Barr virus (EBV) association has been detected in almost 7% of all cases of GC.  However, the EBVaGC specific clinical diagnosis and therapy is not well developed.  So, the research of endoscopic diagnosis and therapy of EBVaGC is required.’  

2   Our communication may have some clinical impact.  ‘Further clinical research in the field of surgery and chemotherapy is required.’ 

Hideo Yanai, MD, PhD.

Chief, Department of Clinical Research,

Hofu Institute of Gastroenterology, Japan. 

Reviewer 2 Report

In the Communications" Endoscopic Diagnosis and Therapy for EBVaGC" Yani et al. recognized EBVaGC as a mainly depressed type with STM-like protrusion in the upper part of the stomach near gastric mucosal atrophic border or remnant stomach using upper GI endoscopy. They claim the recognition rate of 21.4% using GI endoscopy. Since, the manuscript is not well written I am having hard time to comprehend the importance and significance of this work. Thus, I recommend rejecting this manuscript in its current form.

Manuscript is poorly written so I can not understand the significance of their work.

Author Response

Dear Sir,

Our responses to the Reviewer’s comments are as follows. 

To the Reviewer 2,

We added the following sentence

‘Gastric cancer (GC) is one of major health problems throughout the world. Recently, Epstein-Barr virus (EBV) association has been detected in almost 7% of all cases of GC.  However, the EBVaGC specific clinical diagnosis and therapy is not well developed.  So, the research of endoscopic diagnosis and therapy of EBVaGC is required.’  

Hideo Yanai, MD, PhD.

Chief, Department of Clinical Research,

Hofu Institute of Gastroenterology, Japan. 

Reviewer 3 Report

The authors made an interesting study entitled " Endoscopic Diagnosis and Therapy for Epstein-Barr Virus-Associated Gastric Cancer". 

The writers looked into the significance and workings of upper gastrointestinal endoscopy. They also noted that EBVaGC was primarily of the depressed type and had an SMT-like protrusion in the upper section of the stomach, close to the border of atrophic gastric mucosa or remnant stomach. Interesting discovery The selection of patients who might benefit from chemotherapy or endoscopic treatment may depend on the endoscopic diagnosis of EBVaGC.Therefore, I recommend the manuscript for Minor revision.

The manuscript is well written and there are some main points need to be considered before publication.

I recommend the manuscript for acceptance and further publication.

Minor comments:

1. There are some typos throughout the manuscript, please change accordingly.

2. Please include scale bar in all the images.

Language is Fine

Author Response

Dear Sir,

Our responses to the Reviewer’s comments are as follows. 

To the Reviewer 3, 

We included scale bar in the microscopic images.  

These is no scale bar in the endoscopic images. 

Hideo Yanai, MD, PhD.

Chief, Department of Clinical Research,

Hofu Institute of Gastroenterology, Japan. 

Round 2

Reviewer 1 Report

The manuscript can be accepted in current format.

Reviewer 2 Report

None

Fine.